# Acoustic meta-atom with experimentally verified maximum Willis coupling

Anton Melnikov [1,2,3,4], Yan Kei Chiang[4], Li Quan[5], Sebastian Oberst [3], Andrea Alù [5,6], Steffen Marburg[1] & David Powell[4]

Acoustic metamaterials are structures with exotic acoustic properties, with promising applications in acoustic beam steering, focusing, impedance matching, absorption and isolation. Recent work has shown that the efficiency of many acoustic metamaterials can be enhanced by controlling an additional parameter known as Willis coupling, which is analogous to bianisotropy in electromagnetic metamaterials. The magnitude of Willis coupling in a passive acoustic meta-atom has been shown theoretically to have an upper limit, however the feasibility of reaching this limit has not been experimentally investigated. Here we introduce a meta-atom with Willis coupling which closely approaches this theoretical limit, that is much simpler and less prone to thermo-viscous losses than previously reported structures. We perform two-dimensional experiments to measure the strong Willis coupling, supported by numerical calculations. Our meta-atom geometry is readily modeled analytically, enabling the strength of Willis coupling and its peak frequency to be easily controlled.

[1] Vibroacoustics of Vehicles and Machines, Technical University of Munich, Garching b. Munich 85748, Germany. [2] SBS Bühnentechnik GmbH, Dresden 01259, Germany. [3] Centre for Audio, Acoustics and Vibration University of Technology Sydney, Sydney, NSW 2007, Australia. [4] School of Engineering and Information Technology, University of New South Wales, Canberra, ACT 2612, Australia. [5] Department of Electrical and Computer Engineering, The University of Texas at Austin, Austin, TX 78712, USA. [6] Photonics Initiative, Advanced Science Research Center City University of New York, New York, NY 10031, USA. Correspondence and requests for materials should be addressed to A.M. (email: anton.melnikov@tum.de) or to D.P. (email: david.powell@adfa.edu.au)

Acoustic metamaterials[1,2] have demonstrated unique material properties which do not exist naturally, such as negative bulk modulus[3] and negative dynamic mass density[4]. These properties have enabled the development of acoustic superlenses[5–7], barriers[8], cloaking devices[9], and the enhancement of non-linear effects[10]. Metamaterials are typically arrays of sub-wavelength structures, known as meta-atoms, with geometry engineered to control their dynamic mass and stiffness. It has recently been shown that more efficient metamaterial designs can be created by incorporating an additional degree of freedom, represented by the Willis coupling parameter.

Willis coupling is a term in the acoustic and elastic constitutive relations that couples potential and kinetic energy[11–13], analogous to the bianisotropy parameter in electromagnetism[14,15]. The Willis coupling and bianisotropy parameters can only be non-zero in structures with low symmetry[16], for example, in one dimension a structure lacking mirror symmetry in the propagation direction is required[17]. The inclusion of these terms into the constitutive relations has been shown to resolve violations of causality and passivity in metamaterial homogenization[18,19]. Recent work has demonstrated that the incorporation of Willis coupling or bianisotropy into metamaterial structures of sub-wavelength thickness, known as metasurfaces, can improve their efficiency when refracting at large angles[20–22]. While bianisotropy has been demonstrated and engineered in a wide range of electromagnetic meta-atom designs[23], approaches for controlling the degree of Willis coupling in acoustic meta-atoms are not well-established.

Recently, a bound on the maximum value of the Willis coupling parameter was derived, based on the conservation of energy[21]. It was shown how meta-atoms can be designed to reach this theoretical bound, using space-coiling structures with long meander-line channels[21,24–26]. While such structures are advantageous for achieving resonance in a sub-wavelength volume, they are difficult to manufacture reproducibly, typically requiring additive manufacturing techniques. Moreover, their channel widths are often comparable to the viscous and thermal boundary layer thickness, and their channel lengths are of the order of the wavelength, leading to high thermo-viscous losses, and a significant reduction in scattering efficiency[27–29]. It was shown numerically in ref. [21] that the thermo-viscous losses in space-coiling meta-atoms can reduce their Willis coupling magnitude to be significantly lower than the theoretical bound.

Experimental evidence of Willis coupling has been reported in both one-dimensional[2,17] and two-dimensional metamaterial structures[20]. It has also been shown how the effective medium properties of a bulk metamaterial incorporating Willis coupling can be derived from the polarizability of individual acoustic meta-atoms[30]. However, the theoretical limit on the strength of Willis coupling has not been tested experimentally, and it remains unknown how closely this limit may be approached in practice. To resolve this question we propose an acoustic meta-atom, which is designed to realize maximum Willis coupling and to minimize thermo-viscous losses. Experimental results obtained from a fabricated sample are compared to numerical calculations, showing good agreement of the resonant frequency and line-shape, with a reduction in magnitude that we attribute primarily to the energy leakage through the top and bottom waveguide plates, which are not perfectly hard. The simplicity of the structure enables us to present an analytical model for its polarizability, showing how the Willis coupling can be tailored to have any value between zero and the theoretical bound.

## Results

**Meta-atom design.** Acoustic wave interaction with a meta-atom is conveniently described by its polarizability tensor. Due to the sub-wavelength size of meta-atoms, their scattering is dominated by monopole and dipole moments, and their polarizability can be written as

$$\begin{bmatrix} M \\ \mathbf{D} \end{bmatrix} = \boldsymbol{\alpha} \begin{bmatrix} \breve{p}^{\text{inc}} \\ \breve{\mathbf{v}}^{\text{inc}} \end{bmatrix} = \begin{bmatrix} \alpha^{pp} & \boldsymbol{\alpha}^{pv} \\ \boldsymbol{\alpha}^{vp} & \boldsymbol{\alpha}^{vv} \end{bmatrix} \begin{bmatrix} \breve{p}^{\text{inc}} \\ \breve{\mathbf{v}}^{\text{inc}} \end{bmatrix}, \tag{1}$$

where $M$ is the scalar monopole moment, $\mathbf{D}$ is the vector dipole moment, $\boldsymbol{\alpha}$ is the polarizability tensor, $\breve{p}^{\text{inc}}$ and $\breve{\mathbf{v}}^{\text{inc}}$ are the incident pressure and the velocity at the center of the meta-atom[21]. The off-diagonal terms $\boldsymbol{\alpha}^{pv}$ and $\boldsymbol{\alpha}^{vp}$ represent the Willis coupling between the dipolar and monopolar responses.

We show how the reported structures exhibiting Willis coupling[2,17,21] can be replaced by a simpler, and more reproducible structure, by avoiding thin channels and large areas of fluid-structure interfaces. The structure can be tuned to achieve any value of Willis coupling up to the theoretical bounds and can be readily analyzed using a closed form analytical solution which assists in understanding the physical mechanisms behind its operation.

Achieving a strong acoustic polarizability in a small volume requires a resonant structure. Inspired by resonant sonic crystals[8] and Helmholtz resonators with multiple apertures[31–33], our novel meta-atom design is presented in Fig. 1. This meta-atom exhibits Willis coupling due to the asymmetrical neck openings. The air within each neck is treated as an incompressible mass, while the air in the cavity acts as a spring, together creating an oscillating system excited by an incident acoustic wave. As the oscillation occurs in the fluid domain only, this avoids wave coupling through fluid-structure interfaces. Since the structure avoids long and thin channels, thermo-viscous losses are expected to be greatly reduced.

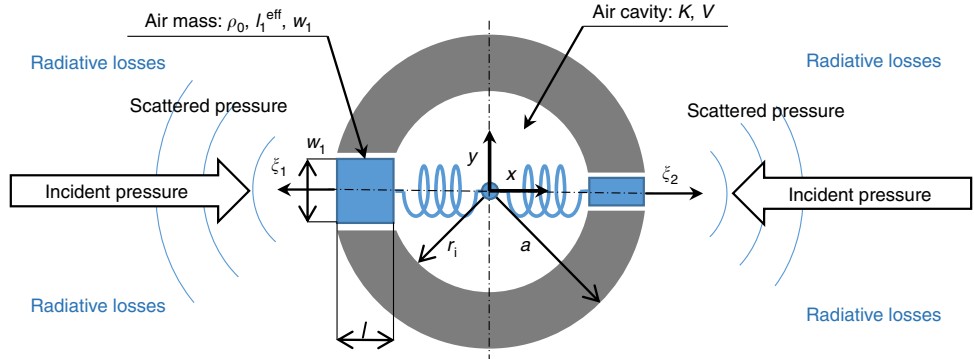

**Fig. 1** Meta-atom geometry and dimensions. Here $a$ is the cylinder radius and $r_i$ is the cavity radius. The neck widths $w_n$ are in general different for each aperture, the neck length $l$ is common to all apertures and the cavity volume $V = \pi r_i^2$

Peak Willis coupling is expected to occur close to the Helmholtz resonator's eigenfrequency, which is dependent on the air mass moving within the apertures and the inner cavity volume. In two dimensions (2D), the moving mass is determined by the aperture cross-sections $A_n = w_n$ and the neck length $l$, which is equal for all apertures due to the inner and outer cylindrical boundaries being concentric. The 2D-volume $V = \pi r_i^2$ is determined by the inner radius $r_i = a - l$, where $a$ is the outer radius. By neglecting radiation damping, the eigenfrequency for $N$ apertures can be approximated as (see Supplementary Note 3)

$$\omega_0 = \frac{c}{r_i} \sqrt{\frac{\sum_{n=1}^{N} w_n}{\pi l}}. \quad (2)$$

A more accurate model including radiation damping is developed subsequently in this work. Generally, an arbitrary number of apertures may be included. However, for controlling Willis coupling along one axis, two oppositely arranged apertures are sufficient, as illustrated in Fig. 1. The apertures can be used to adjust the resonant frequency and the level of Willis coupling. The maximum Willis coupling magnitude is exhibited when the structure is shown in Fig. 1 becomes maximally asymmetric, consistent with the single aperture meta-atoms shown in ref. [21].

**Experimental verification.** To experimentally demonstrate Willis coupling in the presented meta-atom, its polarizability is determined in a 2D experiment. A single aperture meta-atom having maximum asymmetry is manufactured and investigated with the dimensions $a = 20$ mm, $r_i = 10$ mm, and $w = 12$ mm (see Fig. 2a). The incident and scattered pressure fields are measured in a 2D parallel-plate waveguide[29] (see Fig. 2b) with subsequent extraction of the polarizability tensor as detailed in Supplementary Note 5.

The polarizability tensor defined in Eq. (1) has elements with different units and values differing by many orders of magnitude. Therefore it is convenient to introduce the normalized polarizability tensor $\boldsymbol{\alpha}'$, the elements of which are shown for our meta-atom structure in Fig. 2d–g. The normalized values are defined as $\alpha'_{pp} = -2\alpha^{pp}$, $\alpha'_{pv} = \frac{-\sqrt{2}}{\rho c}\alpha^{pv}$, $\alpha'_{vp} = ik\sqrt{2}\alpha^{vp}$, and $\alpha'_{vv} = \frac{ik}{\rho c}\alpha^{vv}$. [21] As required by reciprocity[21], the tensor satisfies $\boldsymbol{\alpha}' = \boldsymbol{\alpha}'^{T-}$, since the off-diagonal terms are equal to each other with a sign reversal (see Fig. 2e, f). The error bars in Fig. 2d–g show the standard deviation of the experimentally extracted polarizability terms, obtained from the least squares fit (see Methods section). To verify that the magnitude of Willis coupling is close to the theoretical maximum, Fig. 3 shows the numerically and analytically calculated $\left|\alpha'_{pv}\right|$ and the experimentally determined $\left|\alpha'_{pv}\right|$ and $\left|\alpha'_{vp}\right|$. For the numerical analysis, the 2D Boundary Element Method (BEM) has been applied. Here, the peak Willis coupling at $k \cdot a = 0.75$ achieves 90% of the theoretical bound $4\omega^{-2}$.

**Polarizability theory.** To show how the Willis coupling and other polarizability components can be tailored by adjusting the meta-atom geometry, we develop a polarizability theory. The predictions of this theory are shown in Figs. 2 and 3. It is based on multiple aperture Helmholtz resonator dynamics for the outward directed particle displacement $\xi_n$ within each aperture (see Supplementary Note 3)

$$i\omega^3 c_n^{rad} \frac{\rho_0 A_n}{c} \xi_n + \omega^2 \rho_0 l_n^{eff} \xi_n - \frac{K}{V}\sum_m A_m \xi_m = p_n^{ext} \quad (3)$$

Here $p_n^{ext}$ is the external pressure at aperture $n$, $\rho_0$ is the medium density, $c$ is the speed of sound, $A_n = w_n$ is the cross-section of the aperture, $K = c^2 \rho_0$ is the bulk modulus, $V$ is the inner cavity volume, $l_n^{eff} = l + c_n^{eff}$ is the effective neck length (including a radiative correction) and $c_n^{rad}$ is the radiative loss coefficient (see Fig. 1). The $\omega^3$-term corresponds to radiative losses, the $\omega^2$-term represents Newton's second law applied to the air in the neck, and the summation over $m$ accounts for coupling between apertures via compression of the center cavity using Hooke's law. Considering all apertures results in a matrix

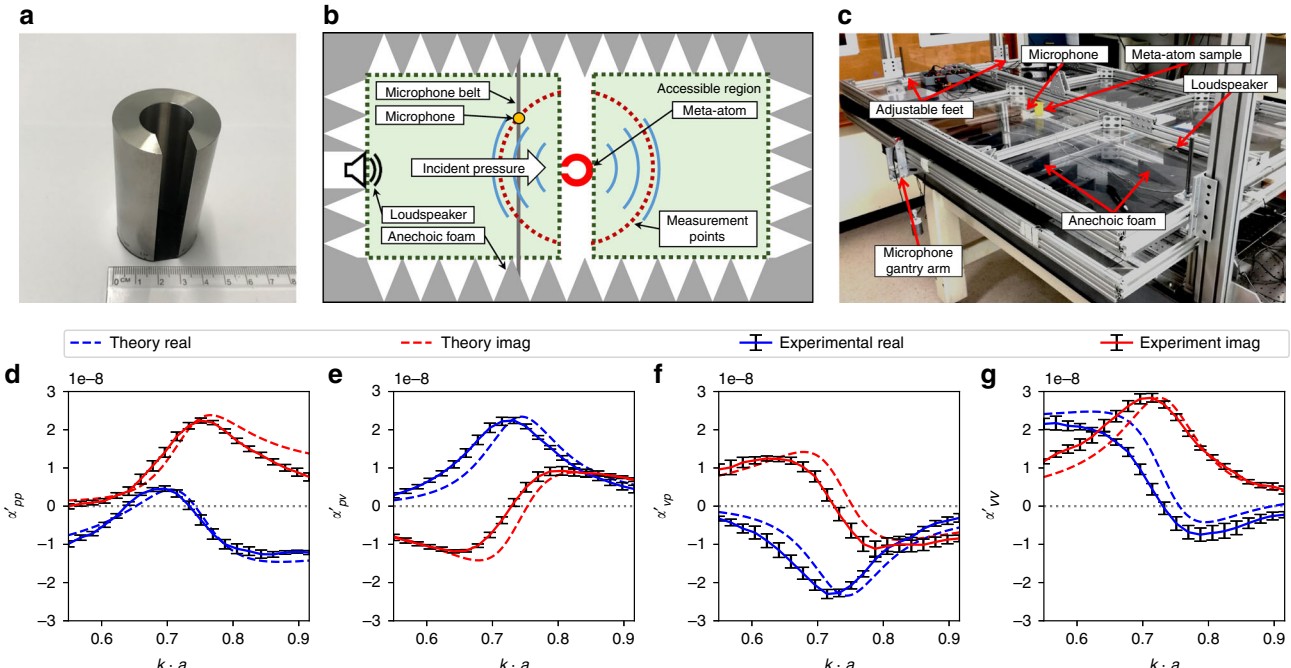

**Fig. 2** Experimental system and results. **a** Stainless steel sample: single aperture meta-atom with $a = 20$ mm, $r_i = 10$ mm, $w = 12$ mm, and $h = 66$ mm. **b, c** Schematic and photograph of the experimental set-up. **d–g** Theoretically predicted and experimentally determined components of the normalized polarizability tensor. The error bars show the standard deviation resulting from the least squares fit over 12 incident angles

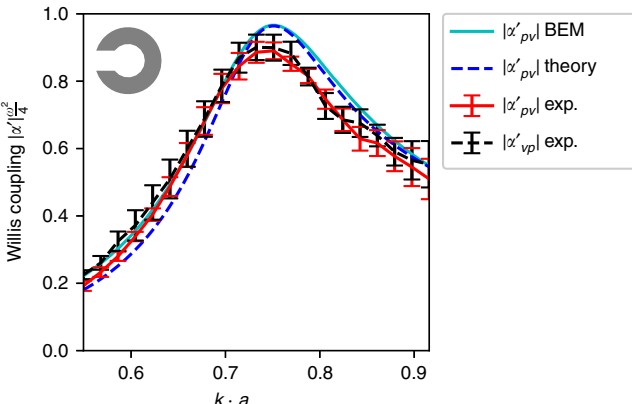

**Fig. 3** Willis coupling magnitude. Result for a single aperture meta-atom obtained from BEM simulation (cyan solid line), polarizability theory (blue dashed line) and experimentally (red solid and black dashed lines), normalized to the theoretical bound $4\omega^{-2}$.

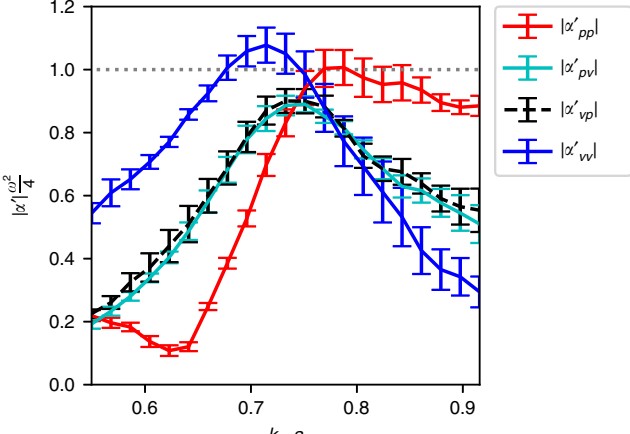

**Fig. 4** Experimental polarizability. Amplitudes of experimentally determined polarizability components showing shared magnitude closely to $k \cdot a = 0.75$

equation $\mathbf{K}_{eq}\boldsymbol{\xi} = \mathbf{p}^{ext}$, where $\mathbf{K}_{eq}$ is the dynamic stiffness matrix given by Supplementary Equation (38). This matrix allows the displacement to be solved for an arbitrary incident pressure field and hence, the contribution of the oscillating air masses (see Fig. 1) to the scattering of the meta-atom. See Supplementary Note 3 for full details of the derivation.

We consider now the shape illustrated in Fig. 1. This restricts the Willis coupling to a single axis and simplifies the problem to a $2 \times 2$ matrix. In this case, the resonator polarizability tensor $\boldsymbol{\alpha}^{res}$ is obtained as

$$\boldsymbol{\alpha}^{res} = \rho_0 \begin{bmatrix} A_1 & A_2 \\ x_1 A_1 & x_2 A_2 \end{bmatrix} \mathbf{K}_{eq}^{-1} \begin{bmatrix} 1 & -ia\rho_0\omega \\ 1 & ia\rho_0\omega \end{bmatrix}, \quad (4)$$

The ratio of aperture widths $\frac{w_1}{w_2} = \frac{A_1}{A_2}$ determines the strength of Willis coupling and can be used to optimize the structure. Expanding Eq. (4) for a single aperture meta-atom with $w_2 = 0$ leads to the following expression for the polarizability

$$\boldsymbol{\alpha}^{res} = \frac{\rho_0 \begin{bmatrix} A_1 & -ia\rho_0\omega A_1 \\ x_1 A_1 & -ia\rho_0\omega x_1 A_1 \end{bmatrix}}{i\omega^3 c^{rad} \frac{\rho_0 A_1}{\pi c} + \omega^2 \rho_0 l_1^{eff} - \frac{K}{V} A_1}. \quad (5)$$

The dynamic stiffness matrix becomes a scalar equation with the remaining projection matrix being singular. The singularity arises because both the monopole and dipole moments are determined from the air movement within a single aperture, but it presents no computational difficulties, since the inverse of the polarizability tensor is not required.

As the Helmholtz resonator is embedded within a cylinder (see Fig. 1), there is an additional influence on the meta-atom polarizability due to the background scattering from the cylinder. Considering only the dipole and monopole terms, the polarizability tensor of a cylinder of radius $a$ is (see Supplementary Note 4)

$$\alpha^{cyl} = \begin{bmatrix} \frac{4}{ik^2 c^2} \frac{J_1(ka)}{H_1^{(1)}(ka)} & 0 \\ 0 & \frac{8\rho_0}{k^3 c} \frac{J_1'(ka)}{H_1^{(1)'}(ka)} \end{bmatrix}, \quad (6)$$

where $J_n$ is a Bessel function, $H_1^{(1)}$ is a Hankel function of the first kind and $k = \omega/c$ is the wavenumber. As expected for a symmetrical geometry, the off-diagonal terms corresponding to Willis coupling are zero. Since the resonator and cylinder are superimposed, they influence the effective incident fields of each other through an additional scattered term:

$$\begin{aligned} \breve{p}_{cyl}^{inc} &= \breve{p}^{inc} + \breve{p}_{res}^{scat} \\ \breve{p}_{res}^{inc} &= \breve{p}^{inc} + \breve{p}_{cyl}^{scat}. \end{aligned} \quad (7)$$

This results in a coupled formulation for the monopole and dipole moments of the cylinder and the Helmholtz resonator as

$$\begin{bmatrix} M^{cyl} \\ D^{cyl} \\ M^{res} \\ D^{res} \end{bmatrix} = \begin{bmatrix} \mathbf{I} & -\boldsymbol{\alpha}^{cyl}\mathbf{E} \\ -\boldsymbol{\alpha}^{res}\mathbf{E} & \mathbf{I} \end{bmatrix}^{-1} \begin{bmatrix} \boldsymbol{\alpha}^{cyl}\mathbf{u}^{inc} \\ \boldsymbol{\alpha}^{res}\mathbf{u}^{inc} \end{bmatrix}. \quad (8)$$

where $\mathbf{u}^{inc} = \begin{bmatrix} \breve{p}^{inc} & \breve{v}_x^{inc} \end{bmatrix}^T$ is the incident field vector and $\mathbf{E} = \mathrm{diag}\left(-\frac{ik^2 c^2}{4}H_0^{(1)}(ka), -\frac{k^2 c}{4a\rho_0}H_1^{(1)}(ka)\right)$ represents the acoustic propagation from the cylinder to each of the apertures. Finally, adding the monopole and dipole moments from Eq. (8) results in

$$\begin{bmatrix} M^{tot} \\ D^{tot} \end{bmatrix} = \begin{bmatrix} M^{cyl} + M^{res} \\ D^{cyl} + D^{res} \end{bmatrix} = \boldsymbol{\alpha}^{tot}\mathbf{u}^{inc}, \quad (9)$$

which gives the total polarizability tensor $\boldsymbol{\alpha}^{tot}$. The components of $\boldsymbol{\alpha}^{tot}$ for a single aperture meta-atom with $a = 20$ mm, $r_i = 10$ mm, $w = 12$ mm, $c = 343$ m·s$^{-1}$, and $\rho_0 = 1.2$ kg·m$^{-3}$ are shown together with the experimental results in Fig. 2.

## Discussion

It has been shown based on passivity and reciprocity conditions[21] that maximum Willis coupling can be achieved only if the polarizability components share the same magnitude and the meta-atom has no losses. Figure 4 shows the magnitude of the experimentally extracted polarizability components, where a crossing point can be observed at $k \cdot a = 0.75$. At this point the magnitudes are very close to each other, giving a precise indication of the frequency of maximum Willis coupling.

The lossless condition cannot be perfectly satisfied, since any fluid medium exhibits thermal and viscous losses, which are greatly magnified close to the boundaries. Furthermore, our two-dimensional parallel-plate waveguide shown in Fig. 2c will leak some energy through the top and bottom plates, since they can only approximate the perfectly hard boundary condition. Although these losses are not treated in our theory, Fig. 2d–g shows that the experimental line-shape and resonant frequency are well described by our theoretical model. To illustrate how closely our meta-atom approaches the theoretical bound, the

magnitude of the Willis coupling is plotted in Fig. 3. The slight frequency shift observable in Fig. 2d–g between the experimental and theoretical values can be attributed to the thermo-viscous losses of air. To investigate these mechanisms within the meta-atom, we model the thermo-viscous losses using the Finite Element Method (FEM), with the resulting Willis coupling shown in Fig. 5. The results reveal a downshifting of the frequency by 2.2% and show that thermo-viscous losses lead to a reduction of 0.32% in the magnitude of Willis coupling. We note that this reduction in magnitude is much less than that previously reported for space-coiling meta-atoms with thin channels, which was ~21% at $k \cdot a \approx 0.75$ (see Supplementary Material of ref. [21]). In Fig. 5 we also plot the Willis coupling magnitude for the space-coiling meta-atom from ref. [21] where the much higher radiative quality factor of the space-coiling structure leads to higher internal dissipation, hence a much greater impact of thermo-viscous losses on the Willis coupling.

In addition to the experimentally demonstrated maximum Willis coupling, the structure presented in Fig. 1 can be tailored to have Willis coupling from zero up to the theoretical bound. To demonstrate this property, normalized Willis coupling for four different parameter sets is shown in Fig. 6. The red meta-atom on top illustrates the single aperture configuration similar to the experimentally investigated structure from Fig. 2a. Once a second aperture with neck width $w_2$ is introduced, Willis coupling is significantly reduced (Fig. 6 magenta and violet lines). This would

result in a frequency shift, as expected from Eq. (2). To avoid this, $w_1$ is tuned to match the peak frequency of the single aperture case. When $w_2$ is further increased and the shape starts to converge to the symmetrical case (Fig. 6 blue, $\frac{w_1}{w_2} \approx 1.1$), the Willis coupling becomes very weak and disappears completely when $\frac{w_1}{w_2} = 1$. This behavior is of practical importance, since it allows tailoring of the Willis coupling to any desired values. A full parametric analysis of the influence of $w_1$ and $w_2$ on the resonant frequency and peak Willis coupling is presented in Supplementary Note 6.

In conclusion, we introduced and experimentally validated a novel meta-atom exhibiting strong Willis coupling and providing a low radiative Q factor. The experiment revealed a Willis coupling magnitude reaching ~90% of the theoretical bound. In this structure, thermo-viscous losses in air are quite small, whereas they are much stronger in previously reported space-coiling meta-atoms. Additionally, the simple shape of our structure facilitates manufacturing and enables accurate analytical modeling. Combining the models of a Helmholtz resonator and a cylindrical scatterer, a theory was developed and shown to agree well with numerical simulations. Since our structure enables Willis coupling to be tailored, this theory can be used to engineer Willis coupling for specific applications.

## Methods

**Extraction of polarizability tensor.** The polarizability relationship given in Eq. (1) is used to illustrate Willis coupling, where it appears as the off-diagonal terms $\boldsymbol{\alpha}^{pv}$ and $\boldsymbol{\alpha}^{vp}$. An extraction method is necessary to obtain the polarizability of a scatterer from an experiment or numerical model. For simplicity, only the 2D case is considered. We build on the method for extracting polarizability from highly symmetric 2D structures in ref. [29] which does not account for Willis coupling. This method makes use of the incident and scattered pressure fields around the object and fits Bessel and Hankel functions to them as $p_{inc}(r, \theta) = \sum_n \beta_n J_n(kr)e^{in\theta}$ and $p_{scat}(r, \theta) = \sum_n \gamma_n H_n^{(1)}(kr)e^{in\theta}$. Since only monopole and dipole components are of importance, the problem can be reduced to considering $n = -1, 0, 1$ terms. Following this, the incident pressure at the meta-atom center is $\check{p}^{inc} = \beta_0$ and the particle velocity $\check{v}_{x,y}^{inc} = \frac{\beta_1 \mp \beta_{-1}}{2c\rho_0}$. The monopole moment $M = \frac{-4\gamma_0}{ik^2 c^2}$ and the dipole moments $D_{x,y} = \frac{-4(\gamma_1 \mp \gamma_{-1})}{ik^3 c^2}$ can be retrieved from the scattered field. For further information see Supplementary Note 1 and 2.

The expansion coefficients $\beta_n$ and $\gamma_n$ can be obtained from measured or numerically determined pressure on circles with radii $R^{inc}$ and $R^{scat}$ respectively. From the orthogonality of exponential functions, the coefficients can be found as

$$\beta_n = \frac{1}{2\pi J_n(kR^{inc})} \int_0^{2\pi} p^{inc}(R^{inc}, \theta)e^{-in\theta}d\theta \tag{10}$$

and

$$\gamma_n = \frac{1}{2\pi H_n^{(1)}(kR^{scat})} \int_0^{2\pi} p^{scat}(R^{scat}, \theta)e^{-in\theta}d\theta. \tag{11}$$

To avoid the singularity of Eq. (10) due to zeros of the Bessel function[29], we ensure $R^{inc} < \frac{2.4}{k}$. However, $R^{scat}$ should be significantly larger than the meta-atom outer radius to reduce near field contributions. These conflicting requirements mean that $R^{inc}$ and $R^{scat}$ are different in general.

To fully determine the polarizability in 2D, $\check{p}^{inc}$, $\check{\mathbf{v}}^{inc}$, $M$ and $\mathbf{D}$ must be determined for at least 3 incident angles. The incident field quantities for all available angles $\theta_{1..m}$ are arranged in a matrix $\boldsymbol{\Upsilon}$ as

$$\boldsymbol{\Upsilon} = \begin{bmatrix} \check{p}^{inc}(\theta_1) & \check{v}_x^{inc}(\theta_1) & \check{v}_y^{inc}(\theta_1) \\ \vdots & \vdots & \vdots \\ \check{p}^{inc}(\theta_m) & \check{v}_x^{inc}(\theta_m) & \check{v}_y^{inc}(\theta_m) \end{bmatrix} \tag{12}$$

Knowing $M$ and $\mathbf{D}$ for each $\theta_{1..m}$, allows the polarizability tensor $\alpha$ to be determined by inversion of $\boldsymbol{\Upsilon}$. For increased robustness, we take additional angles. The polarizability tensor is then determined via least squares as

$$\boldsymbol{\alpha} = (\boldsymbol{\Upsilon}^T\boldsymbol{\Upsilon})^{-1}\boldsymbol{\Upsilon}^T \begin{bmatrix} M(\theta_1) & D_x(\theta_1) & D_y(\theta_1) \\ \vdots & \vdots & \vdots \\ M(\theta_m) & D_x(\theta_m) & D_y(\theta_m) \end{bmatrix}. \tag{13}$$

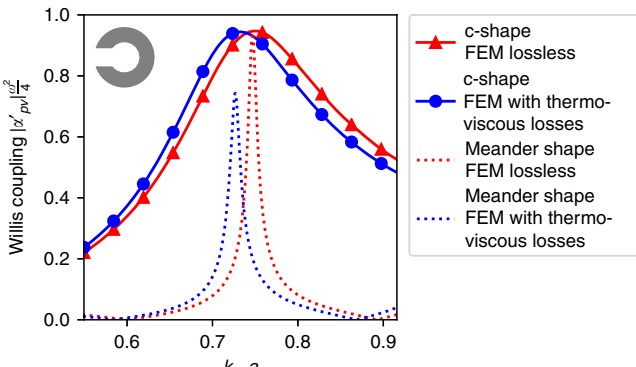

**Fig. 5** Sensitivity to thermo-viscous losses. Numerical comparison of Willis coupling showing the influence of thermo-viscous losses in air, which cause a reduction in magnitude of only 0.32% for the c-shape meta-atom, whereas for the space-coiling structure reported in ref. [21] the magnitude drops by ~21%

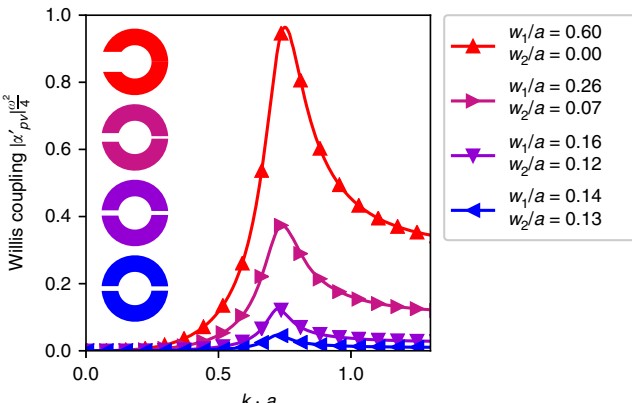

**Fig. 6** Control of Willis coupling. Willis coupling of four different meta-atom geometries, where $w_2$ is varied to control Willis coupling and $w_1$ is tuned to keep the peak frequency fixed

**Numerical model**. To obtain a numerical solution for the polarizability a custom 2D BEM code is used, as described in ref. [29]. It uses continuous elements with quadratic interpolation functions[34] and discretization by collocation method[35] with an adaptive integration scheme[36]. Initially the solids are treated as acoustic hard boundaries. Thermo-viscous losses in air are not included in this formulation. The incident field is a unit intensity plane wave, therefore $\breve{p}^{inc}$ and $\breve{v}^{inc}$ are known explicitly. The medium density and the speed of sound are set to $\rho_0 = 1.2 \, \text{kg} \cdot \text{m}^{-3}$ and $c = 343 \, \text{m} \cdot \text{s}^{-1}$.

To calculate the influence of thermo-viscous losses in air on the polarizability of the meta-atom (Fig. 5) 2D FEM calculations are performed with the COMSOL Multiphysics Thermoviscous Acoustics Module. The acoustic boundary of the meta-atom is treated as rigid. For the lossless case, the mechanical boundary condition of the structure is set to be slip and the thermal boundary is set to be adiabatic. Both the viscosity coefficient and the thermal conductivity coefficient are set as zero in the simulation. For the lossy case, the mechanical boundary is set to be no slip and the thermal boundary is set to be isothermal. The thermal and viscous coefficients used for air are $\rho_0 = 1.2043 \, \text{kg} \cdot \text{m}^{-3}$, $\mu = 1.814 \times 10^{-5} \, \text{Pa} \cdot \text{s}$, $\mu_B = 1.0884 \times 10^{-5} \, \text{Pa} \cdot \text{s}$, $k = 0.025768 \, \text{W} \cdot \text{m}^{-1} \cdot \text{K}^{-1}$, $C_p = 1005.4 \, \text{J} \cdot \text{kg}^{-1} \cdot \text{K}^{-1}$, $\alpha_p = 0.0034112 \, \text{K}^{-1}$, $\gamma = 1.4$ and $\beta_T = 9.8692 \times 10^{-6} \, \text{Pa}^{-1}$. The finite element type is a triangular element with quadratic interpolation functions. The maximum element size is set to approximately 28 elements per wavelength at 2500 Hz.

**Waveguide scattering experiment**. To experimentally determine the incident and scattered pressure fields we use the 2D anechoic waveguide chamber presented in ref. [29]. The propagation medium is air with an acoustic velocity of $343 \, \text{m} \cdot \text{s}^{-1}$. The height of the chamber (66 mm) supports only a single propagation mode at frequencies up to 2600 Hz, making it a 2D wave propagation system. The excitation source is a speaker, excited by a continuous wave with frequency varying between 1500 Hz and 2500 Hz in 50 Hz steps. The response is measured by a microphone, which is moved in two axes by belts driven by stepper motors.

The sample is a single aperture meta-atom shown in Fig. 1a. This meta-atom was manufactured from stainless steel with precisely machined surface with maximum surface roughness of 4 μm (Rz4). Additionally, rubber seals (black) are glued on top and bottom to prevent air leakage from the resonator cavity and to achieve more homogeneous clamping. The incident field is measured at a radius $R^{inc} = 40 \, \text{mm}$. The scattering of the sample is measured at 12 different incident angles (0°, 30°, 60°, 90°, 120°, 150°, 180°, 210°, 240°, 270°, 300°, 330°) at a radius of $R^{scat} = 200 \, \text{mm}$

## Data availability

All relevant data that support the findings of this study are available from the corresponding author upon request.

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

## Acknowledgements

A.M. acknowledges funding supported by German Federal Ministry for Economic Affairs and Energy under the index ZF4128201AT5. A.M. also acknowledges the financial support provided by S.O. through the UTS Centre for Audio, Acoustics and Vibration (CAAV) international visitor funds. D.P. acknowledges funding from the Australian Research Council through Discovery Project DP150103611. A.A. and L.Q. were supported by the National Science Foundation and the Simons Foundation. The authors acknowledge Murat Tahtali from UNSW Canberra for useful discussions on 3D printing and for manufacturing the 3D-printed initial prototype of the single aperture meta-atom.

## Author contributions

A.M. and D.P. developed the analytical model, D.P. designed the experimental set-up; A.M. conceived the meta-atom structure, and designed the experimental sample, validated the results experimentally, numerically and analytically and wrote the paper. Y.K.C. designed the experimental sample and performed measurements to maximize the experimental performance. L.Q. performed the FEM calculations and supported the numerical and analytical study. S.M. and A.A. provided guidance on numerical and theoretical aspects of the study, respectively. A.M., S.O. and D.P. conceptualized the study. D.P. provided guidance on all aspects of the work. All authors contributed in editing the draft.

## Additional information

**Competing interests:** The authors declare no competing interests.

