## [Peer Review File · Nature Communications]

Reviewers' Comments:

Reviewer #1:

Remarks to the Author:

This paper "Acoustic meta-atom with maximum Willis coupling" claims an acoustic meta-atom reaching to the experimental maximum of Willis coupling. When compared to the previous study on the theoretical (lossless) maximum of acoustic bianisotropy (Phys. Rev. Lett. 120, 254301), in this paper authors support their claim by comparing the experimental and numerical results, not deriving the theoretical maximum (under the presence of loss) of Willis coupling.

For the experiment, authors propose and use a meta-atom structure consisting of Helmholtz resonator with two cylinders, in order to reduce thermo-viscous losses. The experimentally achieved (and numerically fitted) 74% of the lossless theoretical-maximum Willis coupling is claimed as the experimentally-obtainable maximum value, while the major penalty is attributed to the material absorption of 3D printed, PLA wall of an imperfect hard boundary.

Considering the recent interest in Willis coupling with potential applications, experimentally achieving / verifying the bound of maximum Willis coupling will be of value in the fields of acoustic metamaterials. However, authors should provide more concrete analysis to support their claims: critically, the relations between the thermo-viscous loss, imaginary surface admittance of viscoelastic materials, and theoretically (more than numerical) achievable maximum Willis coupling (under the presence of above losses). We hereby detail the points of concerns / improvements as listed below.

As described in [Phys. Rev. Lett. 120, 254301], the theoretical limit of Willis coupling is the result of passivity and reciprocity conditions. One achieves the maximum Willis coupling if and only if 1) absolute values of all polarizability being the same, and 2) the acoustic scatterer has no losses.

1). The first condition implies that the maximum bound of Willis coupling can be achieved only when monopolar and dipolar resonances are precisely controlled. However, in Fig.2 (d,g) and also in Fig.S4 (a,d), it is not clear whether α'_{pp} , α'_{pv} , α'_{vp} , and α'_{vv} share the same magnitude of $4/\omega^2$ (4.62×10^{-8} at 1480Hz in Fig.2, and 2.53×10^{-7} at $ka = 0.58$ in Fig.S4). Instead, it seems that α'_{pp} and α'_{vv} values deviate from those of α'_{pv} , α'_{vp} . This implies that, even in lossless (model, numerical) cases, the proposed structure does not fully satisfy the condition of maximum Willis coupling. Authors need to comment on above concerns, possibly with additional data.

2). As well, regarding the loss, it should be carefully claimed whether the (experimentally & numerically) demonstrated "74% of the theoretical maximum" equally implies: 1) proof of "theoretical" maximum under the presence of loss, and 2) proof of achievable experimental maximum irrespective of the use of other materials having lower surface admittance (equivalently, a proof of no higher values of experimental Willis coupling in the future).

It is claimed that most of the acoustic wave dissipation in meander structure results from the thermo-viscous losses, and the authors proposed a new design of a meta-atom having lower thermo-viscous losses. However, it is not clear whether there indeed exists a significant difference between the proposed- and meander- structure. For example, Fig.S2 (c,d) in PRL [Li Quan, PRL 120, 254301] show that the reduction of the peak polarizabilities under the existence of thermo-viscous loss, especially near $ka = 1$ (Fig.S2 c), is not significant, and not much different from the Fig.4 in the submitted manuscript. It is required to provide more quantitative comparison, on the magnitude of the thermo-viscous loss for those two structures. This will strengthen authors' claims of lower loss-structure, as well as the effect of thermo-viscous loss in the Willis coupling operation.

Assuming low thermo-viscous loss in the proposed structure, authors then numerically justified most of the Willis coupling drop (97.2% to 74%) to the imaginary surface admittance. However, it

is not clearly verified whether it is indeed reliable to ignore the additional absorption from the FDM 3D printing (possible surface softness, roughness, etc.). To clarify this issue, it might be possible to use meta-atom samples made of metals (Al or brass), or SLA 3D printing, of lower losses. (Equivalently, in the numerical test, the impact of using different admittance values will be interesting. Absorption of 2.25% resulting in the large ~23% drop in Willis coupling, it implies that there exist a possibility of achieving larger Willis coupling (than demonstrated 74% of theoretical bound), with material of lower absorption < 2.25%). As well, it would be educational if authors could compare numerical (under the influence of surface admittance) and experimental result of α'_{pp} , and α'_{vv} for the consistency check.

To summarize, authors

1. proposed an acoustic meta-atom structure, of lower thermo-viscous loss, and
2. numerically and experimentally compared their result, to claim that
3. the proposed structure provides maximum value of Willis coupling.

However,

1. It is not clear (quantitatively) whether the proposed structure achieved much reduced thermo-viscous loss, when compared to the meander structure.
2. Authors' claim on the role of surface admittance in the reduction of Willis coupling is based on the simple comparison between the experiment and parameter fitting, lacking theoretical justification.
3. There is need of careful claim for the maximum Willis coupling, especially in their title, when other (meta-atom) geometries and (hard-boundary) materials are not rigorously tested yet - leaving the possibilities of higher values exceeding 74% of theoretical bound.

Reviewer #2:

Remarks to the Author:

The work presents a practical realization of an acoustic meta-atom consisting of a cylindrical cavity with asymmetric apertures for controlling Willis coupling. Following a similar approach than the proposed in ref [19], the authors study the Willis coupling in a single meta-atom using polarizabilities. The authors present an analytical formulation of the meta-response and the experimental verification showing agreement with the theoretical predictions. The manuscript is well written and the ideas are clearly presented. A complete analysis of the meta-atoms topology for different configuration shows the versatility of the proposed meta-atom. The supplementary material contains the necessary information for understanding the derivations presented in the main text. In what follows, I give a list of technical comments that could help to improve the current version of the manuscript.

1. In the second paragraph of the introduction, the authors write: "The Willis coupling and bianisotropy parameters can only be non-zero in structures which lack mirror symmetry about one of their axes, ...". In general, this statement is not completely correct. Lack of mirror symmetry means that a mirror image of a particle cannot be superimposed onto the original one by any operations of rotation and translation. In electromagnetism, such bianisotropic particles are called chiral. However, bianisotropy also includes omega-particles that have mirror symmetry. Under this definition, the meta-atoms proposed in this work present mirror symmetry. I suggest clarifying what they mean with mirror symmetry.
2. In order to stress the sub-wavelength size of the proposed metaatom, it will be good to express the parameters in terms of the wavelength at the operation frequency.
3. In page 3 (first paragraph), the author should include the dimensions when they discuss Fig2 (d-g) and Fig. 3. Currently, the dimensions are only in the caption of Fig2.
4. There is a misprint in the vertical axis in Fig 3. The Willis coupling magnitude is defined as $\text{abs}(\alpha'_{pv})$ in the text, but it is expressed as $\text{abs}(\alpha)$ in the figure. This comment also applies to Fig. 5.
5. The derivation and the physical meaning of Eq. (3) are not straightforward. It will be helpful for the reader to refer the supplementary materials and explain the physical meaning when the

equation is introduced.

6. In Fig. S4 (Supplementary materials), the caption seems to be not well defined. The description of (a-d) corresponds with (e-h). Is it so?

7. In the section methods, the authors mention that thermo-viscous losses in the air are modeled using COMSOL. It will be useful for the reader to have more information about the simulation setup.

8. In this work, the authors only analyze the scattering produced by single particles. Could the authors show how this approach can be extended to infinite arrays of met-atoms?

9. Considering that the idea of "maximum Willis coupling" was already introduced and discussed in ref. [19], I recommend that the authors should specify that this work is the experimental demonstration in the title.

The main strength of this work is that the proposed topology may simplify the design and reduce the losses of acoustic devices that require Willis coupling. The analytical formulation could help to develop a systematic and accurate design methodology for this kind of metamaterials. However, this work only deals with the analysis of individual meta-atoms and the potential applicability is not demonstrated with real examples. With the current version, it seems that the work is another implementation of the idea proposed in ref [19]. Therefore, I highly recommend showing the applicability of the proposed idea.

Reviewer #1 (Remarks to the Author):

This paper “Acoustic meta-atom with maximum Willis coupling” claims an acoustic meta-atom reaching to the experimental maximum of Willis coupling. When compared to the previous study on the theoretical (lossless) maximum of acoustic bianisotropy (Phys. Rev. Lett. 120, 254301), in this paper authors support their claim by comparing the experimental and numerical results, not deriving the theoretical maximum (under the presence of loss) of Willis coupling.

For the experiment, authors propose and use a meta-atom structure consisting of Helmholtz resonator with two cylinders, in order to reduce thermo-viscous losses. The experimentally achieved (and numerically fitted) 74% of the lossless theoretical-maximum Willis coupling is claimed as the experimentally-obtainable maximum value, while the major penalty is attributed to the material absorption of 3D printed, PLA wall of an imperfect hard boundary.

Considering the recent interest in Willis coupling with potential applications, experimentally achieving / verifying the bound of maximum Willis coupling will be of value in the fields of acoustic metamaterials. However, authors should provide more concrete analysis to support their claims: critically, the relations between the thermo-viscous loss, imaginary surface admittance of viscoelastic materials, and theoretically (more than numerical) achievable maximum Willis coupling (under the presence of above losses). We hereby detail the points of concerns / improvements as listed below.

As described in [Phys. Rev. Lett. 120, 254301], the theoretical limit of Willis coupling is the result of passivity and reciprocity conditions. One achieves the maximum Willis coupling if and only if 1) absolute values of all polarizability being the same, and 2) the acoustic scatterer has no losses.

The requirements that the absolute values of polarizability are the same and that the acoustic scatterer has no losses are now included into the first paragraph of the Discussion section.

1). The first condition implies that the maximum bound of Willis coupling can be achieved only when monopolar and dipolar resonances are precisely controlled. However, in Fig.2 (d,g) and also in Fig.S4 (a,d), it is not clear whether α'_{pp} , α'_{pv} , α'_{vp} , and α'_{vv} share the same magnitude of $4/\omega^2$ (4.62×10^{-8} at 1480Hz in Fig.2, and 2.53×10^{-7} at $ka = 0.58$ in Fig.S4). Instead, it seems that α'_{pp} and α'_{vv} values deviate from those of α'_{pv} , α'_{vp} . This implies that, even in lossless (model, numerical) cases, the proposed structure does not fully satisfy the condition of maximum Willis coupling. Authors need to comment on above concerns, possibly with additional data.

In the revised manuscript we present results for an improved geometry, fabricated from stainless steel. We have also added a new figure (Fig. 4), where all experimentally extracted polarizability magnitudes have been superimposed. This figure shows that these magnitudes are very close to each other at $ka=0.75$, consistent with our measured Willis coupling closely approaching the theoretical maximum.

2). As well, regarding the loss, it should be carefully claimed whether the (experimentally & numerically) demonstrated “74% of the theoretical maximum” equally implies: 1) proof of “theoretical” maximum under the presence of loss, and 2) proof of achievable experimental maximum irrespective of the use of other materials having lower surface admittance (equivalently, a proof of no higher values of experimental Willis coupling in the future).

In the revised manuscript, we present results achieving 90% of the theoretical maximum. This is achieved by utilising a stainless steel sample with precisely machined surface with maximum surface roughness of $4\mu\text{m}$ (Rz4).

As the remaining losses are much higher than can be accounted for by thermo-viscous losses (see answer below), they are almost certainly due to other sources such as imperfections of the waveguide boundaries, or losses in the rubber seals required to achieve good contact between the meta-atom and waveguide.

Given the complexity of quantifying and modelling these loss mechanisms, it does not seem feasible to prove the impossibility of achieving better results. Instead, our manuscript demonstrates that real metamaterial structures can exhibit Willis coupling close to the theoretical maximum value, meaning that this theoretical value is of practical significance.

It is claimed that most of the acoustic wave dissipation in meander structure results from the thermo-viscous losses, and the authors proposed a new design of a meta-atom having lower thermo-viscous losses. However, it is not clear whether there indeed exists a significant difference between the proposed- and meander- structure. For example, Fig.S2 (c,d) in PRL [Li Quan, PRL 120, 254301] show that the reduction of the peak polarizabilities under the existence of thermo-viscous loss, especially near $ka = 1$ (Fig.S2 c), is not significant, and not much different from the Fig.4 in the submitted manuscript. It is required to provide more quantitative comparison, on the magnitude of the thermo-viscous loss for those two structures. This will strengthen authors' claims of lower loss-structure, as well as the effect of thermo-viscous loss in the Willis coupling operation.

To confirm that our structure has significantly reduced sensitivity to thermo-viscous losses, we have added Figure 5, where we also reproduce the earlier results from Quan et al (2018). As we focus on subwavelength scatterers, the peaks below $ka = 1$ are of main interest, and we compare results for $ka=0.75$. Both sets of results are produced using the same FEM software, with and without the incorporation of thermo-viscous effects. For the structure previously reported by Quan et al, the Willis coupling drops by 21% when thermo-viscous losses are introduced, whereas for our new structure the reduction is only 0.32%. The much lower sensitivity of our new structure to thermo-viscous losses is now discussed explicitly in the Discussion section of the revised manuscript.

Assuming low thermo-viscous loss in the proposed structure, authors then numerically justified most of the Willis coupling drop (97.2% to 74%) to the imaginary surface admittance. However, it is not clearly verified whether it is indeed reliable to ignore the additional absorption from the FDM 3D printing (possible surface softness, roughness, etc.). To clarify this issue, it might be possible to use meta-atom samples made of metals (Al or

brass), or SLA 3D printing, of lower losses. (Equivalently, in the numerical test, the impact of using different admittance values will be interesting. Absorption of 2.25% resulting in the large ~23% drop in Willis coupling, it implies that there exist a possibility of achieving larger Willis coupling (than demonstrated 74% of theoretical bound), with material of lower absorption < 2.25%). As well, it would be educational if authors could compare numerical (under the influence of surface admittance) and experimental result of α'_{pp} , and α'_{vv} for the consistency check.

As per the reviewer's suggestion, in the revised manuscript we report new results for a sample fabrication from stainless steel, with maximum surface roughness of $4\mu\text{m}$ and much better approximation of a hard boundary. This significantly improved the experimentally observed Willis coupling to 90% of the theoretical maximum. As this revised structure addresses both the softness and roughness which may contribute to the finite surface admittance, we no longer incorporate this into our numerical analysis or discussions.

To summarize, authors

1. proposed an acoustic meta-atom structure, of lower thermo-viscous loss, and
2. numerically and experimentally compared their result, to claim that
3. the proposed structure provides maximum value of Willis coupling.

However,

1. It is not clear (quantitatively) whether the proposed structure achieved much reduced thermo-viscous loss, when compared to the meander structure.

The newly introduced Figure 5 clearly demonstrates the reduced sensitivity to thermo-viscous losses.

2. Authors' claim on the role of surface admittance in the reduction of Willis coupling is based on the simple comparison between the experiment and parameter fitting, lacking theoretical justification.

As the new sample made from stainless steel, these surface admittance effects are no longer significant. This parameter fitting is removed from the revised manuscript.

3. There is need of careful claim for the maximum Willis coupling, especially in their title, when other (meta-atom) geometries and (hard-boundary) materials are not rigorously tested yet - leaving the possibilities of higher values exceeding 74% of theoretical bound.

In our revised manuscript, we now achieve 90% of the theoretical bounds, after testing 3 different geometries (all optimal according to numerical results) in both PLA and stainless steel. We have presented strong experimental evidence that the theoretical bounds can be closely approached, which we argue is sufficiently strong to justify our title. We note that the title has been modified as per the suggestion of the second reviewer.

Reviewer #2 (Remarks to the Author):

The work presents a practical realization of an acoustic meta-atom consisting of a cylindrical cavity with asymmetric apertures for controlling Willis coupling. Following a similar approach than the proposed in ref [19], the authors study the Willis coupling in a single meta-atom using polarizabilities. The authors present an analytical formulation of the meta-response and the experimental verification showing agreement with the theoretical predictions. The manuscript is well written and the ideas are clearly presented. A complete analysis of the meta-atoms topology for different configuration shows the versatility of the proposed meta-atom. The supplementary material contains the necessary information for understanding the derivations presented in the main text. In what follows, I give a list of technical comments that could help to improve the current version of the manuscript.

1. In the second paragraph of the introduction, the authors write: "The Willis coupling and bianisotropy parameters can only be non-zero in structures which lack mirror symmetry about one of their axes, ...". In general, this statement is not completely correct. Lack of mirror symmetry means that a mirror image of a particle cannot be superimposed onto the original one by any operations of rotation and translation. In electromagnetism, such bianisotropic particles are called chiral. However, bianisotropy also includes omega-particles that have mirror symmetry. Under this definition, the meta-atoms proposed in this work present mirror symmetry. I suggest clarifying what they mean with mirror symmetry.

The reviewer is partially correct in this criticism, so we have modified the text to avoid confusing or misleading the reader.

As the reviewer noted, Omega particles have one plane of mirror symmetry. However, an essential feature is that they have one principle axis along which there is no mirror symmetry, and waves propagating along this axis of asymmetry can induce bianisotropic effects.

In the one-dimensional case this lack of mirror symmetry is a sufficient condition to exhibit bianisotropy/Willis coupling (a reference by Muhlstein et al is now cited), and by extension this is sufficient for our experimental case of engineering Willis coupling along a single axis of a two-dimensional structure.

For higher dimensions there is no simple statement that can be made, as various combinations of rotation, inversion and mirror symmetry have different effects on the bianisotropy parameter (to avoid confusion with chirality we note that this specifically means the Omega bianisotropy parameter). For the electromagnetic case a full symmetry analysis is given in the book by Serdyukov et al, so this is cited.

2. In order to stress the sub-wavelength size of the proposed metaatom, it will be good to express the parameters in terms of the wavelength at the operation frequency.

We changed all the figures to be expressed in ka .

3. In page 3 (first paragraph), the author should include the dimensions when they discuss Fig2 (d-g) and Fig. 3. Currently, the dimensions are only in the caption of Fig2.

The dimensions are now included where the Fig. 2 (a) is introduced, in the first paragraph of the Experimental Verification section.

Lines 145 to 148: “A single aperture meta-atom having maximum asymmetry is manufactured and investigated with the dimensions $a = 20\text{mm}$, $r_i = 10\text{mm}$, and $w = 12\text{mm}$ (see Fig. 2(a))”

4. There is a misprint in the vertical axis in Fig 3. The Willis coupling magnitude is defined as $\text{abs}(\alpha'_{pv})$ in the text, but it is expressed as $\text{abs}(\alpha)$ in the figure. This comment also applies to Fig. 5.

Changed in text referring to Fig. 3. Changed the axis label of Fig. 5. The y-axis of Fig. 3 and 4 is still labeled as $\text{abs}(\alpha)$, since there are different polarizability magnitudes in the same plot.

5. The derivation and the physical meaning of Eq. (3) are not straightforward. It will be helpful for the reader to refer the supplementary materials and explain the physical meaning when the equation is introduced.

A reference to supplementary material is added directly in front of the equation. The physical meaning of each terms has been clarified in the subsequent paragraph.

Lines 187 to 191: “The ω^3 -term corresponds to radiative losses, the ω^2 -term represents Newton’s second law applied to the air in the neck, and the summation over m accounts for coupling between apertures via compression of the center cavity using Hooke’s law.”

6. In Fig. S4 (Supplementary materials), the caption seems to be not well defined. The description of (a-d) corresponds with (e-h). Is it so?

This is now corrected.

7. In the section methods, the authors mention that thermo-viscous losses in the air are modeled using COMSOL. It will be useful for the reader to have more information about the simulation setup.

The COMSOL model is now described in the revised Methods section.

Lines 365 to 380:

“To calculate the influence of thermo-viscous losses in air on the polarizability of the meta-atom (Fig. 5) 2D FEM calculations are performed with the COMSOL Multiphysics Thermoviscous Acoustics Module. The acoustic boundary of the meta-atom is treated as rigid. For the lossless case, the mechanical boundary condition of the structure is set to be slip and the thermal boundary is set to be adiabatic. Both, the viscosity coefficient and the thermal conductivity coefficient are set as zero in the simulation. For the lossy case, the mechanical boundary is set to be no slip and the thermal boundary is set to be isothermal. The thermal and viscous coefficients used for air are $\rho_0 = 1.2043 \text{ kg/m}^3$, $c = 343.14 \text{ m/s}$, $\mu = 1.814 \times 10^{-5} \text{ Pa}\cdot\text{s}$, $\mu_B = 1.0884 \times 10^{-5} \text{ Pa}\cdot\text{s}$, $k = 0.025768 \text{ W/m}\cdot\text{K}$, $C_p = 1005.4 \text{ J/kg}\cdot\text{K}$, $\alpha_p = 0.0034112 \text{ K}^{-1}$, $\gamma = 1.4$ and $\beta_T = 9.8692 \times 10^{-6} \text{ Pa}^{-1}$. The finite element type is a triangular element with

quadratic interpolation functions. The maximum element size is set to approximately 28 elements per wavelength at 2500Hz.”

8. In this work, the authors only analyze the scattering produced by single particles. Could the authors show how this approach can be extended to infinite arrays of met-atoms?

Sieck et al 2017 has already shown how the polarizability of an individual meta-atom (as obtained in our work) can be augmented with mutual coupling terms (based on the same polarizability model), to obtain the effective polarizability in a periodic array, and hence obtain effective medium properties. This point is now mentioned in the last paragraph of the introduction.

Lines 66 to 70: “It has also been shown how the effective medium properties of a bulk metamaterial incorporating Willis coupling can be derived from the polarizability of individual acoustic meta-atoms [30].“

Directly applying our method to an infinite array would probably require significant adaptation to base it on a Bloch wave solution instead of a scattering solution.

Sieck, C. F., Alù, A. & Haberman, M. R. Origins of Willis coupling and acoustic bianisotropy in acoustic metamaterials through source-driven homogenization. Physical Review B 96, 104303

9. Considering that the idea of “maximum Willis coupling” was already introduced and discussed in ref. [19], I recommend that the authors should specify that this work is the experimental demonstration in the title.

The title is changed to reflect the experimental character of this paper.

Title: „Acoustic meta-atom with experimentally verified maximum Willis coupling“

The main strength of this work is that the proposed topology may simplify the design and reduce the losses of acoustic devices that require Willis coupling. The analytical formulation could help to develop a systematic and accurate design methodology for this kind of metamaterials. However, this work only deals with the analysis of individual meta-atoms and the potential applicability is not demonstrated with real examples. With the current version, it seems that the work is another implementation of the idea proposed in ref [19]. Therefore, I highly recommend showing the applicability of the proposed idea.

The purpose of this manuscript is to investigate experimentally the maximum bounds on Willis coupling. We argue that this is a significant result, since so far these bounds were only predicted theoretically, and experimental verifications are important. Making use of Willis coupling to develop more complex structures such as metasurfaces is beyond the scope of the present work.

Reviewers' Comments:

Reviewer #1:

Remarks to the Author:

Authors faithfully addressed reviewers' concerns in this revision. Especially, the verification of the authors' claim is now much clearer, with the use of hard-boundary materials for the construction of meta-atom.

One (optional) suggestion remains. In Fig.6 authors numerically analyzed meta-atom structures which are "similar" to the meta-atom used in the experiment (Fig.2a).

I suggest (optional) authors include the numerical result of Fig.2a structure as an overlay to Fig.6, or, to update Fig.6 with the experimental structure used in Fig2a (with double apertures).

Reviewer #2:

Remarks to the Author:

The revisions provided by the authors have satisfied the majority of my concerns, and they have also considerably improved the manuscript. I recommend this work for publication.

Response to reviewer's comments:

Reviewer #1 (Remarks to the Author):

Authors faithfully addressed reviewers' concerns in this revision. Especially, the verification of the authors' claim is now much clearer, with the use of hard-boundary materials for the construction of meta-atom.

One (optional) suggestion remains. In Fig.6 authors numerically analyzed meta-atom structures which are "similar" to the meta-atom used in the experiment (Fig.2a).

I suggest (optional) authors include the numerical result of Fig.2a structure as an overlay to Fig.6, or, to update Fig.6 with the experimental structure used in Fig2a (with double apertures).

We have updated Fig. 6 to consider the experimental sample shape. Furthermore, we have updated Supplementary Figures 3, 4, 5, and 6 to include the experimental structure.

Reviewer #2 (Remarks to the Author):

The revisions provided by the authors have satisfied the majority of my concerns, and they have also considerably improved the manuscript. I recommend this work for publication.

No changes have been requested by this reviewer.